# Antibiotic Resistance and Therapy Outcome in *H. pylori* Eradication Failure Patients

**DOI:** 10.3390/antibiotics9030121

**Published:** 2020-03-13

**Authors:** Saracino I.M., Pavoni M., Zullo A., Fiorini G., Saccomanno L., Lazzarotto T., Cavallo R., Antonelli G., Vaira D.

**Affiliations:** 1Department of Surgical and Medical Sciences, University of Bologna, 40138 Bologna, Italymatteo.pavoni@studio.unibo.it (P.M.); giulia.fiorini@fastwebnet.it (F.G.); laura.saccomanno@studio.unibo.it (S.L.); 2Department of Molecular Medicine, Sapienza University of Rome, 00185 Rome, Italy; guido.antonelli@uniroma1.it; 3Gastroenterology Unit, ‘Nuovo Regina Margherita’ Hospital, 00153 Rome, Italy; zullo66@gmail.com; 4Department of Experimental, Diagnostic and Specialty Medicine, University of Bologna, 40138 Bologna, Italy; tiziana.lazzarotto@unibo.it; 5Microbiology and Virology Unit, University Hospital Città Della Salute e Della Scienza di Torino, 10126 Turin, Italy; rossana.cavallo@unito.it

**Keywords:** *H. pylori*, non-naïve patients, antibiotic resistance, antibiotic susceptibility test, rescue therapies

## Abstract

*Helicobacter pylori* (*H. pylori*) eradication fails in a definite amount of patients despite one or more therapeutic attempts. Curing these patients is progressively more difficult, due to development of antibiotic resistance. Current guidelines suggest testing antibiotic susceptibility in *H. pylori* isolates following two therapeutic attempts. Aim: to evaluate the development of antibiotic resistance, MIC values trends and therapeutic outcomes in patients who failed at least one *H. pylori* eradication therapy. Methods: consecutive patients, referred to perform upper gastrointestinal endoscopy (UGIE) to our Unit from January 2009 to January 2019 following at least one therapeutic attempt were considered. Bacterial resistance towards clarithromycin, metronidazole and levofloxacin was tested. Patients received either a susceptibility-guided therapy or Pylera^®^. Results: a total of 1223 patients were *H. pylori* positive, and antibiotic susceptibility was available for 1037. The rate of antibiotic resistance and MIC values significantly increased paralleling the number of previous therapeutic attempts. Eradication rates of antibiogram-tailored therapies remained stable, except for the sequential therapy if used as a third line. As a rescue treatment, the Pylera^®^ therapy achieved cure rates comparable to those of the other culture-guided therapies. Conclusions: A significant increase in the secondary resistance towards the three tested antibiotics was observed, both as rate and MIC values, in correlation with the number of therapy failures. These findings should be considered when administering an empirical second-line therapy. Pylera^®^ therapy eradication rates are comparable to culture-tailored therapies.

## 1. Introduction

*Helicobacter pylori* (*H. pylori*) infections play a role in different gastrointestinal diseases, such as peptic ulcers, gastric mucosa associated lymphoid tissue lymphoma (MALT), and gastric cancer [1]. Antibiotic resistance is a growing problem for eradication therapies [2]. The selective pressure of the antibiotic intake causes modification in the genetic pattern of *H. pylori* that remains stable [3]. In 2017, the World Health Organization classified clarithromycin resistant *H. pylori* strains as a “high-priority” pathogen [4]. Resistance to fluoroquinolones can also impair the efficacy of eradication regimens [5], whereas resistance to nitroimidazoles can be partially overcome in vivo when used in quadruple therapies [6]. Current Italian and European guidelines suggest either a bismuth quadruple therapy or a levofloxacin triple therapy as second line therapies, whilst treatment should be guided by antimicrobial susceptibility testing after failure of a second-line therapy [5,7,8]. Although there are several factors influencing efficacy of an anti-*H. pylori* therapy [9,10], resistance to antibiotics remains the most relevant [8,10,11,12]. In particular, the double clarithromycin-metronidazole resistance significantly affects the cure rate of standard eradication regimens [10]. In addition, recent studies suggest that not only the susceptible/resistant status can affect therapy outcome, but also the MIC levels (low or high) for resistance [13,14]. This study aimed to evaluate the development of antibiotic resistance, MIC values trends and therapeutic outcomes in patients who failed at least one *H. pylori* eradication therapy.

## 2. Results

Between 2009 and 2019, a total of 1223 patients were diagnosed with *H. pylori* infection following at least one eradication therapy, and antibiotic susceptibility testing was available in 1037 of them. Clinical and demographic characteristics of these patients were provided in Table 1.

Overall, *H. pylori* isolates were resistant towards clarithromycin in 83.1% (95% CI = 80.7–85.2), metronidazole in 66.7% (95% CI = 63.8–69.5), and levofloxacin in 47.2% (95% CI = 44.2–50.2) of cases. The pattern of secondary antibiotic resistance was provided in Table 2.

Resistance rate towards all antibiotics progressively increased in correlation with the number of previous therapeutic failures. In detail, as compared to patients who failed only one therapy, those with two failures had significantly higher resistance rates towards all antibiotics, including double and triple resistance. A further significant increase in the resistance rate towards metronidazole, double clarithromycin-metronidazole and triple resistances was observed in patients who failed three therapies. At the fourth or more therapeutic failure, only resistance towards levofloxacin continued to increase (Table 3).

As far as MIC values towards different antibiotics is concerned, the levels of resistance in *H. pylori* isolates significantly increased following two or more failures as compared to those following only one therapy failure (Figure 1).

Overall, following culture-based therapies the eradication rates at ITT (intention to treat) and PP (per protocol) analysis were 84.2% (95% CI = 76.4–89.8) and 91.4% (95% CI = 84.5–95.4), respectively, with sequential therapy; 75.6% (95% CI = 71.5–79.3) and 83.5% (95% CI = 79.6–86.8) with rifabutin-based therapy; 80.4% (95% CI = 76.2–83.9) and 86.6% (95% CI = 82.8–89.6) with levofloxacin-based regimen. Pylera^®^ therapy eradication rates were 86.8% (95% CI =75.1–93.4) and 90% (95% CI = 79.0–95.7). The eradication rates of different regimens accordingly to the previous therapeutic failures were reported in Table 4. 

## 3. Discussion

There is evidence that antibiotic resistance is increasing in *H. pylori* isolates and, therefore, it is very important to carry out periodic assessments of antibiotic resistance rates and to monitor the efficacy of different treatments [2,15,16,17]. In the present study, we evaluated the rate of secondary resistance towards different antibiotics and its role in the therapy outcome. We observed there was a statistically significant increase in the resistance rates towards clarithromycin, levofloxacin, and metronidazole which correlate with the number of previous treatment failures. 

Similarly, our data found that mean of MIC levels increased significantly following two or more therapeutic attempts. This finding is relevant when considering that some evidences suggest the MIC values affect therapy efficacy, higher the MICs lower the success rate [13]. Therefore, prescribing an inappropriate empirical therapy could trigger a vicious circle that leads to therapy failure, increase in resistance rates, increase in MIC levels and a consequent higher probability of another therapeutic failure [18]. 

Focusing on eradication rates following culture-based rescue therapies, the efficacy of different regimens remained generally stable among patients who failed one or more treatments. Only sequential therapy had a statistically significant decrease in eradication rates when used as third-line therapy. In our study, the Pylera^®^ regimen was prescribed regardless of the resistance pattern, achieving eradication rates comparable to that of tailored therapies, and its efficacy decreased only when administered as fourth-line treatment. This observation suggests that this regimen could be successfully used as second- or third-line regimen without resorting in bacterial culture. Unfortunately, Pylera^®^ regimen is really complex and causes side-effects more frequently than other therapies. Indeed, as many as 14 tablets administered four times for 10 days are needed, and side-effects occurred in more than 30% of cases with a higher rate of earlier therapy interruption [19,20]. Recently, the role of 14 days, high-dose dual therapy with esomeprazole 40 mg and amoxicillin 1 g, both twice a day, has been renewed as a rescue therapy [21]. Therefore, in clinical practice, a judicious proposal could be the use of high-dose dual therapy as second-line therapy and the Pylera^®^ as rescue therapy.

## 4. Materials and Methods

### 4.1. Patients

This was a retrospective, single-center study (Sant’Orsola Hospital, Bologna, Italy) evaluating consecutive Italian patients referred by their physicians to our unit for upper endoscopy. Patients who had previously failed at least one *H. pylori* therapeutic attempt, referred to our Unit from January 2009 to January 2019, were considered. The exclusion criteria were: (1) age <18 years; (2) previous gastric surgery; (3) use of proton pump inhibitors or antibiotics in the 2 weeks before the endoscopy; (4) known allergy to macrolides, nitroimidazoles or penicillins. Each patient provided us personal information such as age, weight, height, smoking habits, daily intake of alcohol, familiarity for gastric cancer, educational qualifications, and region of birth and residence. The number of previous eradication attempts was registered. All participants provided written informed consent. The study was approved by the local ethical Committee and performed according to guidelines for Good Clinical Practice [22] and the Declaration of Helsinki [23]

### 4.2. Endoscopy and H. pylori Assessment 

During endoscopy biopsy specimens (two from the antrum, two from the corpus, one from incisura angularis) were taken for histology (Haematoxylin-eosin for pathological assessment and Giemsa for *H. pylori* staining). One additional antral biopsy was used for bacterial culture and drug-susceptibility test. Based on endoscopic reports, for the purposes of the study, patients with either a peptic ulcer (ulceration 5 mm in diameter) or mucosal erosions (superficial lesion of 4 mm) in the stomach or duodenum were grouped together as “peptic ulcer disease” (PUD). Non-ulcer dyspepsia was diagnosed when no macroscopic lesion were detected at endoscopy and patients were included in “non-ulcer disease” (NUD) group. Treatment success was evaluated by using a standard ^13^C-Urea Breath Test (UBT) performed 6 to 8 weeks after treatment ended. Patients undergoing therapy for fewer than 7 days were considered as drop-outs, and those who did not undergo UBT testing after treatment were considered lost to follow-up evaluation. All patients were required to undergo a visit at the end of therapy to assess both compliance to therapy and the incidence of side effects, evaluated by a direct interview. 

### 4.3. H. pylori Susceptibility Testing

Biopsy specimens collected for bacterial culture were streaked immediately onto commercial selective medium Pylori Agar (BioMérieux Italia S.p.A., Italy). The plates were incubated under microaerobic conditions at 37 °C for 3 to 5 days. Once incubated, the colonies resembling *H. pylori* were identified by oxidase, catalase, and urease tests. Suspensions from the primary plates were prepared in sterile saline solution to McFarland opacity standard 4 (approximately 10^9^ cells/mL) to perform an *E-Test* (BioMérieux Italia S.p.A., Italy). A total of 4 agar plates for every *H. pylori* strains were streaked in 3 directions with a swab dipped into each bacterial suspension to produce a lawn of growth. Three *E-Test* strips (clarithromycin 0.016–256 ug/ml, metronidazole 0.016–256 ug/ml, and levofloxacin 0.008–32 ug/ml) were placed each onto a separate plate, which was incubated immediately in a microaerobic atmosphere at 37 °C for 72 hours. A fourth plate was used as positive control. Clarithromycin, metronidazole and levofloxacin resistance break points for the minimal inhibitory concentration (MIC) are: greater than 0.5 mg/L, greater than 8 mg/L, and greater than 1 mg/L, respectively, according to the updated recommendations of the European Committee on Antimicrobial Susceptibility Testing (EUCAST 2019) [24]. In 2010 EUCAST lowered clarithromycin cut-off from 1 ug/ml to 0.5 ug/ml, strains isolated in 2009 with 0.5 ug/ml< MICs ≤1 ug/ml are still considered susceptible. In 2015 EUCAST established for clarithromycin that 0.25 ug/ml < MICs ≤ 0.5 ug/ml were to be considered as “indeterminate”, suggesting not to administer the drug in this case, so these strains were considered as “resistant” in this study [24]. Drug susceptibility test was not performed for amoxicillin and tetracycline because in Europe the resistance rate is lower than 1% [17].

### 4.4. Therapy Regimens 

Based on antibiotic susceptibility testing, patients received one of the following therapies (named as “culture-based” or “antibiogram-tailored” therapies). Patients without antibiotic resistance received a 10-day sequential therapy with esomeprazole 40 mg and 1000 mg of amoxicillin, both twice daily for 5 days followed by esomeprazole 40 mg, clarithromycin 500 mg and tinidazole 500 mg, all twice daily, for the remaining 5 days. Patients infected with a levofloxacin-susceptible strain (and resistant to clarithromycin and/or metronidazole) were treated with a 10-day triple therapy of esomeprazole 40 mg, levofloxacin 250 mg, and amoxicillin 1 g, all twice daily. Patients with levofloxacin-resistant strains (and resistant to clarithromycin and/or metronidazole) received a 12-day triple therapy with esomeprazole 40 mg and amoxicillin 1 g, both twice daily, and rifabutin 150 mg once daily. Starting from 2016, 10-day Pylera^®^ regimen was prescribed (regardless of the resistance pattern), with three in one capsule containing 140 mg bismuth subcitrate potassium, 125 mg metronidazole and 125 mg tetracycline, administered as 3 tablets four times a day plus esomeprazole 20mg twice a day. To enhance compliance, patients were instructed carefully to adhere to the drug regimen, after they had been given a written synopsis of the daily regimen. They were advised of the possible side-effects. 

### 4.5. Statistical Analysis 

Means and their 95% confidence intervals were calculated as suggested by Newcombe et al. [25]. Comparisons among patient subgroups were performed using the Chi-square test (Yates correction when appropriate). Differences between sample means were analyzed with Student T test. A P level less than 0.05 was considered significant. Statistical analysis was performed with MedCalc19.1.

## 5. Conclusions

A significant increase in the secondary resistance towards the three tested antibiotics was observed, both as rate and MIC values, in correlation with the number of therapy failures. These findings should be considered when administering an empirical second-line therapy. Pylera^®^ therapy eradication rates are comparable to culture-tailored therapies.

## Figures and Tables

**Figure 1 antibiotics-09-00121-f001:**
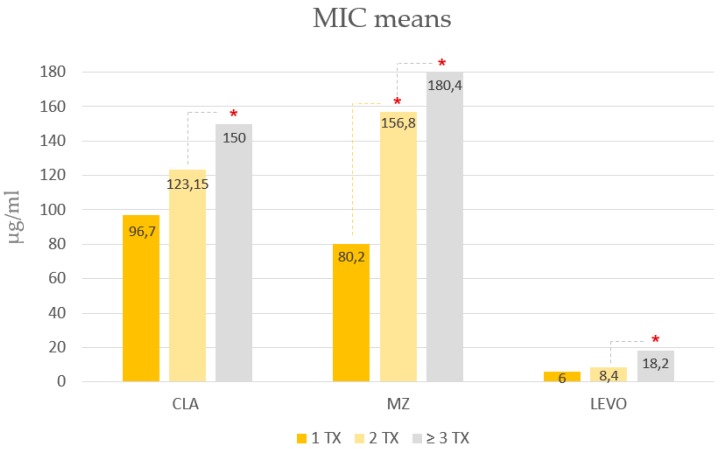
MIC values towards different antibiotics following one or more therapy failures. TX = therapy. Cla: clarithromycin. Mz: metronidazole. Levo: levofloxacin. * *p* < 0.05.

**Table 1 antibiotics-09-00121-t001:** Demographic and clinical characteristics of patients.

Population Features	N.	%	95% CI
**Patients**	1037	−	−
**Male**	335	32.3	29.5–35.2
**Female**	702	67.7	64.8–70.5
**Age mean**	52	−	−
**BMI mean**	24.1	−	−
**Smokers**	182	17.6	15.3–20.0
**Alcohol**	94	9.1	7.4–11.0
**Cardioaspirin**	58	5.6	4.3–7.2
**Familiarity for gastric cancer**	208	20.1	17.7–22.6
**NUD**	943	90.9	89.0–92.6
**PUD**	82	7.9	6.3–9.7
**MALT**	8	0.8	0.3–1.5
**Gastric cancer**	4	0.4	0.1–1.0

95% CI: 95% Confidence interval. BMI: Body Mass Index. NUD: Non-ulcer disease. PUD: Peptic ulcer disease. MALT: Mucosal-associated lymphoid tissue.

**Table 2 antibiotics-09-00121-t002:** Resistance patterns in *Helicobacter pylori* (*H. pylori*) isolates.

Resistance rates (1037 strains)			
Resistance Patterns	No.	%	95% CI
**ClaR MetroR LevoR**	403	38.8	35.9–41.9
**ClaR MetroR LevoS**	237	22.8	20.4–25.5
**ClaR MetroS LevoR**	53	5.1	3.9–6.6
**ClaR MetroS LevoS**	169	16.3	14.1–18.6
**ClaS MetroR LevoR**	26	2.5	1.7–3.6
**ClaS MetroR LevoS**	26	2.5	1.7–3.6
**ClaS MetroS LevoR**	8	0.7	0.3–1.5
**ClaS MetroS LevoS**	115	11.0	9.3–13.1

95% CI: 95% Confidence Interval. Cla: clarithromycin. Mz: metronidazole. Levo: levofloxacin. R: resistant.

**Table 3 antibiotics-09-00121-t003:** Comparison of resistance rates based on the number of therapeutic failures.

N. of failures	ClaR	MzR	LevoR	RR	RRR
	%	%	%	%	%
**1 vs. 2 therapies**	75.7 vs. 85.2 *	52.2 vs. 70.6 *	29.1 vs. 45.8 *	45.7 vs. 65.4 *	19.7 vs. 38.0 *
**2 vs. 3 therapies**	85.2 vs. 90.0 *	70.6 vs. 80.0 *	45.8 vs. 46.0	65.4 vs. 77.0	30.0 vs. 61.5 *
**3 vs. 4 therapies**	90.0 vs. 92.0	80.0 vs. 85.3	46.0 vs. 73.3 *	77.0 vs. 80.0	61.5 vs. 69.3
**4 vs. ≥ 5 therapies**	92.0 vs. 94.5	85.3 vs. 91.9	73.3 vs. 81.0	80.0 vs. 89.1	69.3 vs. 75.6

* *p* < 0.05. R: resistant. Cla: clarithromycin. Mz: metronidazole. Levo: levofloxacin. RR: double clarithromycin-metronidazole resistance. RRR: triple clarithromycin-metronidazole-levofloxacin resistance.

**Table 4 antibiotics-09-00121-t004:** Cure rates accordingly to previous therapeutic failures.

No. of Failures	Therapy	No.	PP	95% CI	ITT	95% CI
**1 (415 pts)**	Sequential	67	98.4	91.5–99.7	92.5	83.7–96.8
	Levofloxacin-based	221	88.5	83.2−92.2	79.7	73.8–84.4
	Rifabutin-based	109	89.0	81.0–93.9	74.3	65.4–81.6
	Pylera^®^	18	94.1	73.0–99.0	88.9	67.2–96.9
**2 (310 pts)**	Sequential	29	84.5 *	71.0–96.0	79.3	71.6–91.1
	Levofloxacin-based	132	88.1	81.3–92.6	84.1	76.9–89.3
	Rifabutin-based	129	82.2	74.3–88.0	75.2	67.1–81.8
	Pylera^®^	20	94.7	75.3–99.1	90.0	69.9–97.2
**≥3 (312 pts)**	Sequential	18	81.2	57.0–93.4	72.2	49.1–87.5
	Levofloxacin-based	63	77.1	65.1–85.8	74.6	72.7–83.7
	Rifabutin-based	216	82.2	76.3–86.8	76.9	70.8–82.0
	Pylera^®^	15	80.0 *	54.8–92.9	80.0	54.8–92.9

* Sequential therapy had a statistically significant (P < 0.05) decrease in eradication rates if used as third line therapy. Pylera^®^ had a statistically significant decrease in eradication rates if used as fourth line therapy. Pts: patients. PP: per protocol analysis. ITT: intention to treat analysis.

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
