# Peer review of "Antibiotic Resistance and Therapy Outcome in H. pylori Eradication Failure Patients"

_antibiotics, 2020, doi:10.3390/antibiotics9030121_

Round 1

Reviewer 1 Report

This is a useful, concise clinical assessment which should have useful applications in therapeutic assessments. 

The article could do with some proof reading. Some minor comments:

Line 20: replace "quote" with "number" or "amount"

Line 54: "infections"

Line 58: H. pylori should be in italics

Line 155: Remove "RR" from table legend since it's not part of this table

Line 246: should be "increasing"

Author Response

Dear Collegue,

Thank you for your letter.

We have followed all your helpfull suggestions.

Please see the following review followed point by point.

Line 20: replace "quote" with "number" or "amount" done

Line 54: "infections" one

Line 58: H. pylori should be in italics done

Line 155: Remove "RR" from table legend since it's not part of this table done

Line 246: should be "increasing" done

Reviewer 2 Report

Saracino and Vaira et al. studied the antibiotic resistance of patients with at least one failed H. pylori eradication therapy. They found the resistance rate and MIC values increased for subsequent therapy treatments. They also compared different therapy methods and suggested some guidelines for second and third-line therapies. This work is valuable for treating patients with H. pylori. However, the presentation of the paper needs to be improved. Especially, this journal has a broader audience not just readers with medical background. The authors should provide more background information for the general readers. Some major issues with this paper are described below.

  1. The numbers from line 213 to 218 are very difficult to find in Table 4. It seems that they don’t match with each other. Please double check and provide a better description.
  2. Define what is culture-based therapies.
  3. How many patient data are in the subsequent therapies? Are there 1037 patient data or only those with eradication failure?
  4. Antibiogram-tailored therapies are mentioned in the abstract, but it appears nowhere in the text.

Some minor points:

  1. Write out the full name of abbreviation the first time it is used in the text. Additionally, avoid using abbreviation that is not frequent in the text. For example, ITT, PP, PPI, t.i.d are confusing.
  2. Align table 2 caption and remove RR, which is not appeared in the table.

Author Response

Dear Collegue,

Thank you for your letter.

We have followed all your helpfull suggestions.

Please see the following review followed point by point.

  1. The numbers from line 213 to 218 are very difficult to find in Table 4. It seems that they don’t match with each other. Please double check and provide a better description.

Numbers in line 213-218 are total eradication rates. They are not reported in table 4. Table 4 reports eradication rates tailored for previous eradication failures.

2. Define what is culture-based therapies.

We added a more clear explanation in chapter 4.4

3. How many patient data are in the subsequent therapies? Are there 1037 patient data or only those with eradication failure?

1037 were the patients referred to perform upper gastrointestinal endoscopy (UGIE) to our Unit from January 2009 to January 2019 following at least one previous therapeutic failure. Number of patients who were prescribed each therapy were added to table 4

4. Antibiogram-tailored therapies are mentioned in the abstract, but it appears nowhere in the text.

Clarified in chapter 4.4

Some minor points:

5.Write out the full name of abbreviation the first time it is used in the text. Additionally, avoid using abbreviation that is not frequent in the text. For example, ITT, PP, PPI, t.i.d are confusing.  

done

6. Align table 2 caption and remove RR, which is not appeared in the table.

done